# One of the Deepest Genera of Antipatharia: Taxonomic Position Revealed and Revised

Tina N. Molodtsova [1,*], Dennis M. Opresko [2], Michael O'Mahoney [2], Ulyana V. Simakova [1], Galina A. Kolyuchkina [1], Yessenia M. Bledsoe [3], Teresa W. Nasiadka [4], Rachael F. Ross [5] and Mercer R. Brugler [2,3,6]

1   Shirshov Institute of Oceanology RAS, 36 Nakhimovsky Prospect, 117218 Moscow, Russia
2   Department of Invertebrate Zoology, National Museum of Natural History, Smithsonian Institution, 10th St. & Constitution Ave. NW, Washington, DC 20560, USA
3   Department of Natural Sciences, University of South Carolina Beaufort, 801 Carteret Street, Beaufort, SC 29902, USA
4   Department of Biochemistry and Cell Biology, Stony Brook University, 100 Nicolls Road, Stony Brook, NY 11794, USA
5   Terra Global Capital, LLC, 6114 La Salle Avenue, Suite 441, Oakland, CA 94611, USA
6   Division of Invertebrate Zoology, American Museum of Natural History, Central Park West at 79th Street, New York, NY 10024, USA
*   Correspondence: tina@ocean.ru

**Abstract:** The genus *Abyssopathes* Opresko, 2002, comprises deep-sea black corals known almost exclusively from lower bathyal and abyssal depths, mainly from seamounts covered by cobalt-rich crusts and areas of polymetallic nodules. The taxonomical position of the genus and its placement in the family Schizopathidae has been repeatedly questioned, but fruitlessly. Known only in extremely deep habitats, these corals have rarely been collected in a state suitable for morphological or molecular studies that could help to clarify their status. Recently, increasing attention has been paid to the study of fauna associated with deep-sea minerals. Using material of *Abyssopathes lyra* (Brook, 1889) sampled during these studies, we transfer the genus *Abyssopathes* from the family Schizopathidae to the family Cladopathidae based on morphological and molecular data. Morphological data includes six mesenteries in the polyps, a unique pinnulation pattern found only in genera within the Cladopathidae, and relatively short polyp tentacles, a feature typical of some cladopathids. Sequencing data, consisting of 626 bp from the mitochondrial *cox1* gene, showed that *Abyssopathes* is 99% identical to *Chrysopathes* Opresko, 2003, *Cladopathes* Brook, 1889, *Heteropathes* Opresko, 2011, and *Trissopathes* Opresko, 2003 (all Cladopathidae), in this gene region.

**Keywords:** black corals; deep-sea; histology; mesenteries; mitochondrial DNA; phylogeny

## 1. Introduction

It is generally considered that species of the order Antipatharia occur primarily in the deep sea, with up to 75% of all hitherto known antipatharian species recorded below 50 m [1,2]. However, this approximation is possibly exaggerated, as many genera of the families Antipathidae, Myriopathidae, and Aphanipathidae are found predominantly at shallow (0–30 m) and mesophotic (30–150 m) depths [3,4] and only up to 31.6% of hitherto described species occur below 800 m [5]. Not all antipatharian families are equally represented in the deep sea. At depths of 800–3500 m, the dominant families are the Cladopathidae, Antipathidae, and Schizopathidae (13.68%, 14.74%, and 51.58%, respectively) [5]. Below 3500 m, the only families present are the Leiopathidae, Cladopathidae, and Schizopathidae, with the Schizopathidae comprising 85.71% of the black coral species known from the abyssal and hadal zones [5].

A close evolutionary relationship between Cladopathidae and Schizopathidae was revealed in the first molecular phylogenies of the order [6]. Both families share the absence

of the *cox1* intron in the mitochondrial genome [7]. The only reliable morphological feature that has been used to date to distinguish these two families is the number of polypar mesenteries: six in Cladopathidae and ten in Schizopathidae. Many features of gross morphology are shared between the two families: general patterns of pinnulation, presence of striatum in monopodial colonies, transversely elongated polyps, etc. The common morphological features of the Cladopathidae and Schizopathidae families make it difficult to accurately position a species that is devoid of polyps or has poorly preserved soft tissues when no histological or molecular methods can be used. It can be illustrated by the situation with *Sibopathes macrospina* Opresko, 1993, which originally was assigned to the family Cladopathidae based on the presence of only six mesenteries in the polyps [8], but molecular data showed that specimens having identical external morphology clustered with the morphologically similar genus *Parantipathes* Brook, 1889, in the family Schizopathidae [6,7,9]. However, until genetic-grade material of *Sibopathes gephura* van Pesch, 1914, the type species of the genus *Sibopathes* van Pesch, 1914, is collected and analyzed, the taxonomic position of the genus *Sibopathes* cannot be resolved [6,7].

The genus *Abyssopathes* Opresko, 2002 (Figure 1), is one of the deepest occurring genera of black corals and is found exclusively at lower bathyal and abyssal depths [3,5]. Although a species of the genus *Schizopathes* Brook, 1889, is known from hadal depths [3,5], in general, the genus *Schizopathes* more commonly occurs in the bathyal zone [5,10]. In contrast, all three hitherto known species of *Abyssopathes* were reported from a similar depth range (3000–5000 m) from oligotrophic deep basins, often in connection with seamounts covered by cobalt-rich crusts and areas of polymetallic nodules [3,11–14]. *Bathypathes lyra* Brook, 1889 (Figures 1 and 2a), the type species of *Abyssopathes*, was originally included by Brook [11] in the genus *Bathypathes* Brook, 1889, subfamily Schizopathinae of the family Antipathidae. When the genus *Abyssopathes* was established [12], it was reclassified within the family Schizopathidae in the absence of any data on polyp structure [11,12]. The taxonomical position of the genus and its placement in the family Schizopathidae has been repeatedly questioned because of similarities in gross morphology to species of the family Cladopathidae (Figures 2 and 3) [11,12]; however, these corals have rarely been collected in a state suitable for histological or molecular studies that could help to clarify their status.

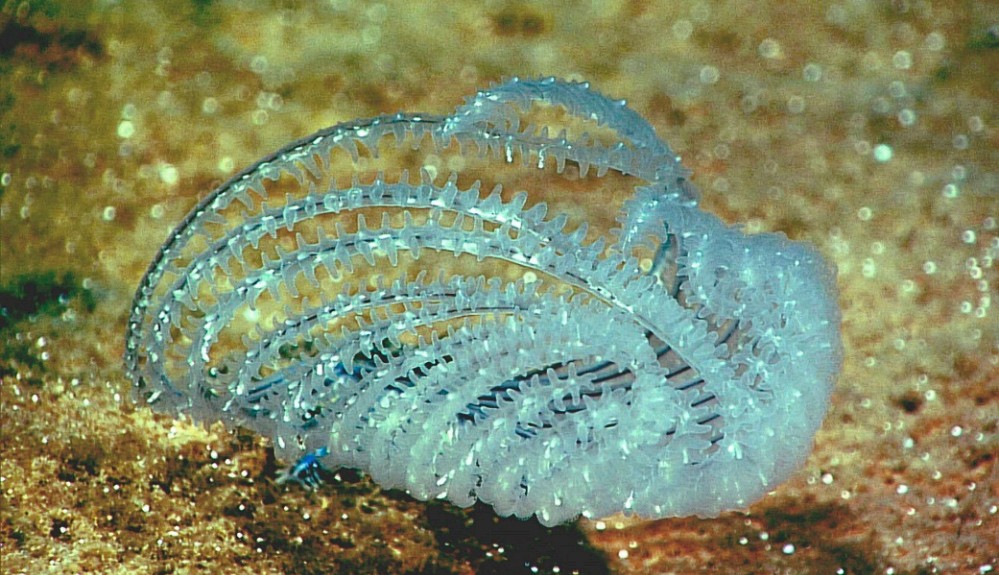

**Figure 1.** *Abyssopathes lyra* (Brook, 1889). In situ photo of specimen USNM 1453760 with a characteristic basket-like corallum. Photo courtesy of the NOAA OER (Note: color enhanced to show details).

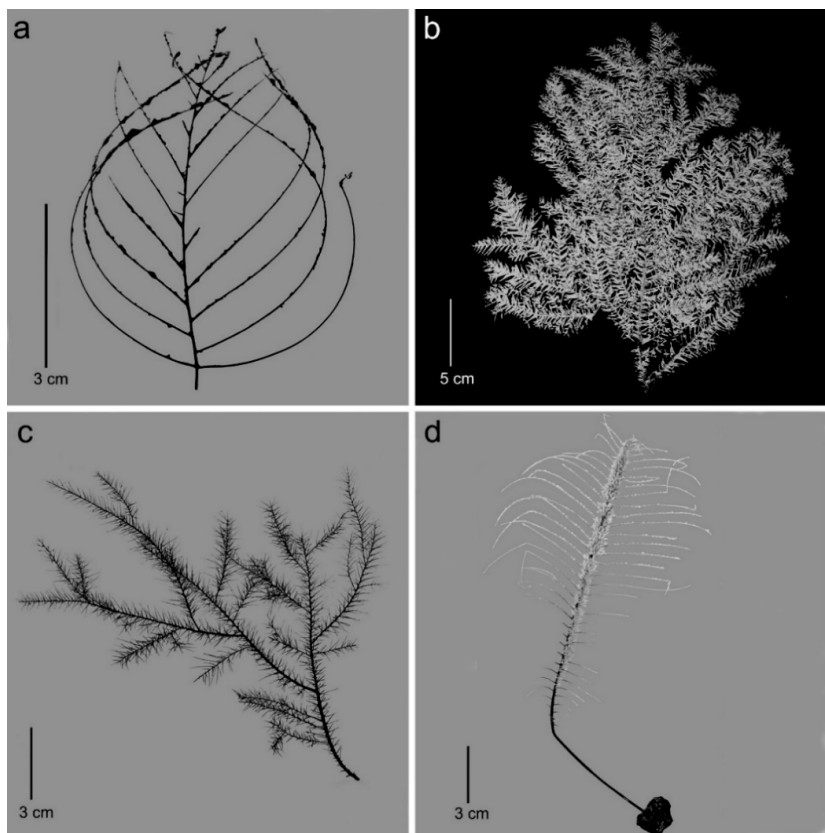

**Figure 2.** Select genera of the family Cladopathidae: (**a**) *Abyssopathes lyra* (Brook, 1889) (syntype, NHMUK 1890-4-9-22); (**b**) *Trissopathes pseudotristicha* Opresko, 2003 (USNM 1070975); (**c**) *Chrysopathes speciosa* Opresko, 2003 (paratype, USNM 92520); (**d**) *Heteropathes pacifica* (Opresko, 2005) (holotype, USNM 1070758).

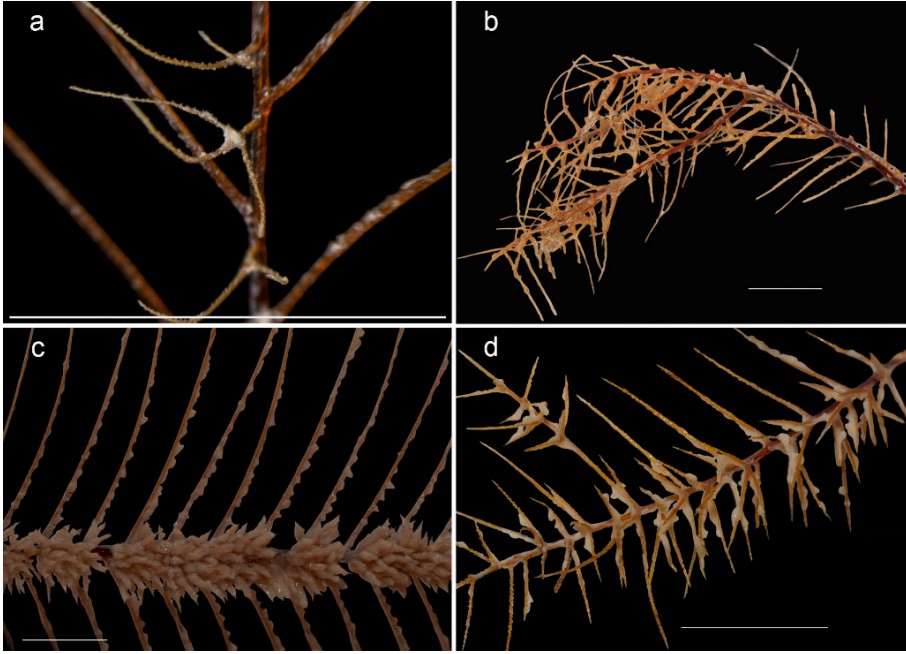

**Figure 3.** Close-ups of select genera of the family Cladopathidae: (**a**) *Abyssopathes lyriformis* Opresko, 2002 (holotype, USNM 83567); (**b**) *Sibopathes gephura* van Pesch, 1914 (holotype ZMA Coel 3283), (**c**) *Heteropathes pacifica* (holotype, USNM 1070758), (**d**) *Trissopathes pseudotristicha* (holotype, RMNH Coel. 32045). Scale 10 mm.

Recently, the National Oceanic and Atmospheric Administration (NOAA) organized and implemented a Pacific-wide field campaign CAPSTONE (Campaign to Address Pacific monument Science, Technology, and Ocean NEeds). A number of previously poorly studied biotopes were investigated and sampled using the dual body, 6000 m-rated, ROV system Deep Discoverer (D2) [15]. On one of the dives to an abyssal habitat off the Phoenix Islands, a specimen of *Abyssopathes lyra* was collected at a depth of 5772.36 m and recovered on deck with the polyps in a reasonably good state of preservation, which allowed for both histological and molecular studies. Below we present our data to clarify the taxonomic position of this abyssal genus.

## 2. Materials and Methods

Morphological characters used to distinguish genera in the families Cladopathidae and Schizopathidae were analyzed based on existing literature [3,5,8,12,16–21] and original data (Table 1, Supplementary Table S1). For terminology, see [6,22].

**Table 1.** Comparison of gross colony features of *Abyssopathes* Opresko, 2002, with other hitherto known genera of Cladopathidae and Schizopathidae [3,5,8,12,16–21].

| Genus | Mature Corallum Growth Form | Striatum | Primary Pinnulation | Typical Number of Rows of Primary Pinnules | Lateral or Posterolateral Pinnules | Anterior or Anterolateral Pinnules |
|---|---|---|---|---|---|---|
| ***Abyssopathes*** | monopodial | present | laterals regular; anteriors irregular | 3 to 4 | simple; basal subopposite others alternating | absent, simple or subpinnulate |
| **CLADOPATHIDAE** | | | | | | |
| *Chrysopathes* | branched | present | regular; in biserial groups | 5 to 6 | subpinnulate; basal subopposite, others alternating | subpinnulate |
| *Cladopathes* | densely branched | no data | Irregular | 4 to 5 | simple;alternating | subpinnulate |
| *Heteropathes* | monopodial | present | laterals regular; anteriors irregular | 3 to 4 | simple; basal subopposite, others alternating | subpinnulate |
| *Hexapathes* | monopodial | present | laterals regular; anteriors irregular | 3 to 4 | simple; basal subopposite others alternating | simple or subpinnulate |
| *Trissopathes* | laxly or densely branched | at least in some species | regular; in biserial groups | 3 to 4 | usually simple; basal subopposite others alternating | subpinnulate |
| *Sibopathes* | laxly branched | present | regular; in biserial groups | 4 to 6 | simple; basal subopposite others alternating | simple |
| **SCHIZOPATHIDAE** | | | | | | |
| *Alternatipathes* | monopodial or rarely branched | absent | regular | 2 | simple; all alternating | absent |
| *Bathypathes* | monopodial; rarely branched | present or absent | regular | 2 | simple; all subopposite or all alternating | absent |
| *Dendrobathypathes* | sparsely to densely branched | no data | regular | 2 | subpinnulate; all alternating | absent |
| *Dendropathes* | densely branched | no data | regular; in biserial groups | 4 | simple; all alternating | simple |
| *Diplopathes* | laxly branched | no data | regular | 2 | simple; all alternating | absent |

**Table 1.** *Cont.*

| Genus | Mature Corallum Growth Form | Striatum | Primary Pinnulation | Typical Number of Rows of Primary Pinnules | Lateral or Posterolateral Pinnules | Anterior or Anterolateral Pinnules |
|---|---|---|---|---|---|---|
| *Lillipathes* | usually laxly branched | present | regular; in biserial groups | 4 | simple; all alternating | simple |
| *Parantipathes* | monopodial or laxly branched | present or absent | regular; in biserial groups | 6 to 12 | simple; all alternating | simple |
| *Saropathes* | monopodial? | no data | regular; in biserial groups | 4 | subpinnulate; all alternating | subpinnulate |
| *Schizopathes* | monopodial | absent | regular | 2 | simple; all alternating | absent |
| *Stauropathes* | sparsely or densely branched | at least in some species | regular | 2 | simple; all subopposite | absent |
| *Taxipathes* | sparsely branched | no data | regular; in biserial groups | 6 | simple; all alternating | simple |
| *Telopathes* | sparsely branched | no data | regular | 2 | simple; alternating or subopposite | absent |
| *Umbellapathes* | monopodial-like stalked with branched crown | absent | regular | 2 | simple or subpinnulate; all alternating | absent |

A colony of *Abyssopathes* (USNM 1453760; subsample BPBM D2466) (Figure 1) was collected by the NOAA Ship *Okeanos Explorer* (NOAA OER) as a part of the CAPSTONE project during cruise EX1703, Dive 16 to the Howland and Baker unit of the Pacific Remote Islands Marine National Monument (PRIMNM) and the Phoenix Islands Protected Area (PIPA). The specimen was preserved in 95% ethanol on board the ship. Dive 16 was to an unnamed hadal trough off the Phoenix Islands (4.38° S, 173° W) at a depth of 5772.36 m. The sampled colony was identified as *Abyssopathes lyra*, the type species of the genus. Whole genomic DNA was extracted from a subsample of this specimen at the Smithsonian Institution, and the 5′-end of the mitochondrial cytochrome c oxidase subunit I (*cox1*) gene was sequenced via traditional Sanger sequencing in both the forward and reverse direction. After trimming low-quality base calls from the ends of each read, 626 base pairs of overlapping sequence remained (GenBank Accession Number MT350286). This 626 bp fragment of *cox1* was added to the *igrC* multiple sequence alignment from [6]. Additionally, schizopathid and cladopathid mitochondrial *cox1* sequences were extracted from the complete mitogenome-based phylogeny presented in [6], as well as a recent paper describing three species within the newly established genus *Diplopathes* Opresko, 2022 [21], and added to the *igrC* alignment, resulting in a total of 88 taxa. The *igrC* alignment was truncated on the 5′ and 3′ end to exactly match the length [626 bp] of the *Abyssopathes lyra* sequence. XSEDE on the CIPRES Science Gateway v3.3 [23] was used to construct a maximum likelihood-based phylogeny using IQ-Tree v2.1.2 with 1000 ultrafast bootstrap replicates and the GTR+I+G model of sequence evolution [24]. The resulting phylogenetic reconstruction was used to infer the evolutionary relationship of *Abyssopathes lyra* to representatives of four genera within the family Cladopathidae (*Chrysopathes* Opresko, 2003, *Cladopathes* Brook, 1889, *Heteropathes* Opresko, 2011, and *Trissopathes* Opresko, 2003) and eleven genera within the family Schizopathidae (*Alternatipathes* Molodtsova and Opresko, 2017, *Bathypathes* Brook, 1889, *Dendropathes* Opresko, 2005, *Dendrobathypathes* Opresko, 2002, *Diplopathes, Lillipathes* Opresko, 2002, *Parantipathes, Saropathes* Opresko, 2002, cf. *Sibopathes* van Pesch, 1914, *Telopathes* MacIsaac and Best, 2013, and *Stauropathes* Opresko, 2002).

The sample identified as cf. *Sibopathes macrospina* in the phylogenetic reconstructions [6,7,9] is morphologically very similar to *Sibopathes*, but was not evaluated histologically or by any other means to determine the number of mesenteries in the polyps.

Genetic distance estimates were obtained using the Kimura 2-Parameter within MEGA X [25]. K2P parameters included uniform rates among sites and pairwise deletion of gaps/missing data.

Polyps of the same specimen (subsample BPBM D2466) were subjected to histological analysis. Two fragments of a pinnule with a polyp were re-hydrated through a graded ethanol series (95–30%), then placed in Carlisle's solution [26,27] and soaked for 24 h. After the Carlisle's solution treatment, the fragments were rinsed and dehydrated through a graded ethanol series (30–95%), followed by: (1) four changes of isopropanol; (2) mixtures isopropanol:mineral oil 5:1 and 2:1; and (3) mineral oil alone. The samples were then infiltrated in two changes of Histomix® paraffin medium (BioVitrum, Bolshoy Prospekt Vasilyevsky Island, 68, lit. A, St. Petersburg, 199106, Russia) at 56 °C. The resulting Histomix® soaked samples were embedded in Histomix ®, blocked, and sectioned at 5 μm. Sections of polyps were mounted on microscopic slides, deparaffinized, re-hydrated, stained with Mayer's hemalum [27], dehydrated, dealcoholized in toluol, and mounted with synthetic mounting media "Bio Mount" (BioVitrum, Russia).

## 3. Results

We have considered morphological, histological, and molecular data in re-evaluating the taxonomic position and relationships of *Abyssopathes*.

### 3.1. Morphological Analysis

Two of the three species in the genus *Abyssopathes* are similar in general appearance to species of Hexapathes Kinoshita, 1910, and *Heteropathes* (Figure 3a,c, Table 1, Supplementary Table S1) in that they have two rows of simple, elongate, lateral primary pinnules, as well as one or more rows of anterior primary pinnules which can be simple or subpinnulate. Furthermore, the pinnulation pattern in which the two lowermost lateral pinnules are subopposite and all the others are arranged alternately is found exclusively in *Hexapathes*, *Heteropathes*, and *Abyssopathes* (Table 1, Supplementary Table S1). In contrast, there are no genera in the family Schizopathidae that have a similar pinnulation pattern. It should be noted that *Abyssopathes anomala* Molodtsova and Opresko, 2017, is uniquely different from the other two species of *Abyssopathes* in that it lacks anterior primary pinnules. The presence of anterior pinnules is a characteristic of all other taxa within the family Cladopathidae, and its absence in *A. anomala* suggests that it is an autapomorphic character state.

In situ photographs of *Abyssopathes lyra* (Figures 1 and 4a) indicate that the tentacles, when fully expanded, are relatively short compared to the transverse diameter of the polyp. A similar condition has also been observed in some in situ photos of the polyps of species of *Heteropathes* (Figure 4b), suggesting a possible shared characteristic; however, actual measurements of the tentacle length of living polyps are not available. In contrast, the tentacles of the polyps in genera in the family Schizopathidae are usually much longer in proportion to the transverse diameter (Figure 4c,d). Polyps of *Abyssopathes*, however, differ from those of other members of the Cladopathidae (*Trissopathes*, *Cladopathes*, *Hexapathes*, and *Heteropathes*) in having a very short oral cone.

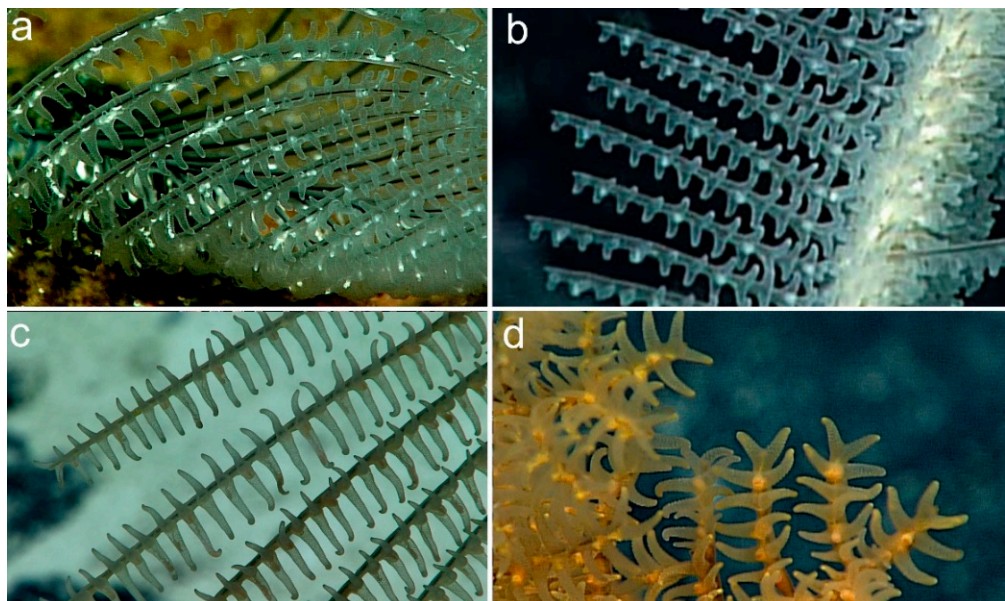

**Figure 4.** Polyps of Cladopathidae and Schizopathidae: (**a**) *Abyssopathes lyra* (USNM 1453760); (**b**) *Heteropathes pacifica*; (**c**) *Bathypathes* cf *patula* Brook, 1889; (**d**) *Stauropathes staurocrada* Opresko, 2002. In situ photos taken by the ROV *Deep Discoverer* (NOAA Ship *Okeanos Explorer*, Cruises EX 1703, EX1708, and EX1504L2). Photos courtesy of the NOAA OER.

*3.2. Histological Analysis*

Carlisle's solution treatment [26,27] of the samples of *Abyssopathes lyra* (BPBM D2466) gave adequate results in softening the scleroproteinaceous axial material of the pinnule, and successful sectioning was carried out without seriously affecting the soft tissue. However, a study of the resulting microscopic slides indicated that the initial preservation of the delicate polyp tissue in 95% ethanol caused the dissolution of the gland cells and probably also some damage to the polyp epidermis (Figure 5a–c). Sections of two polyps of *Abyssopathes lyra* (BPBM D2466) were obtained. Polyps of *A. lyra* demonstrated a few unique morphological features not reported in previous studies of schizopathids or cladopathids [3,8,11,28,29]. Thus, both sectioned polyps were found to have a very thick mesogleal layer in the polyp wall (30–65 μm) and tentacles (20–35 μm). In addition, a cross-sectional view of the upper part of the polyps revealed that the stomodeum has a very distinct Maltese cross or four-leaf outline (Figures 5a,b and 6a). The cross becomes more elongated in a transverse direction lower down in the polyp at a level corresponding to the base of the tentacles (Figure 5a), but it still retains a recognizable four-leaf shape. It should be noted that a stomodeum of a similar but more elongated shape was reported in *Cladopathes plumosa* Brook, 1889 (Figure 6c) [11], although it has not been found in any non-cladopathid taxa.

In all the studied histological sections, only six primary mesenteries and no secondary mesenteries were found (Figure 5). Two mesenteries were connected to each lateral fold of the stomodeum, and one was connected to each transversal fold (Figure 5). The presence of only six mesenteries is the morphological feature that is characteristic of the family Cladopathidae (Figure 6a–c) [16].

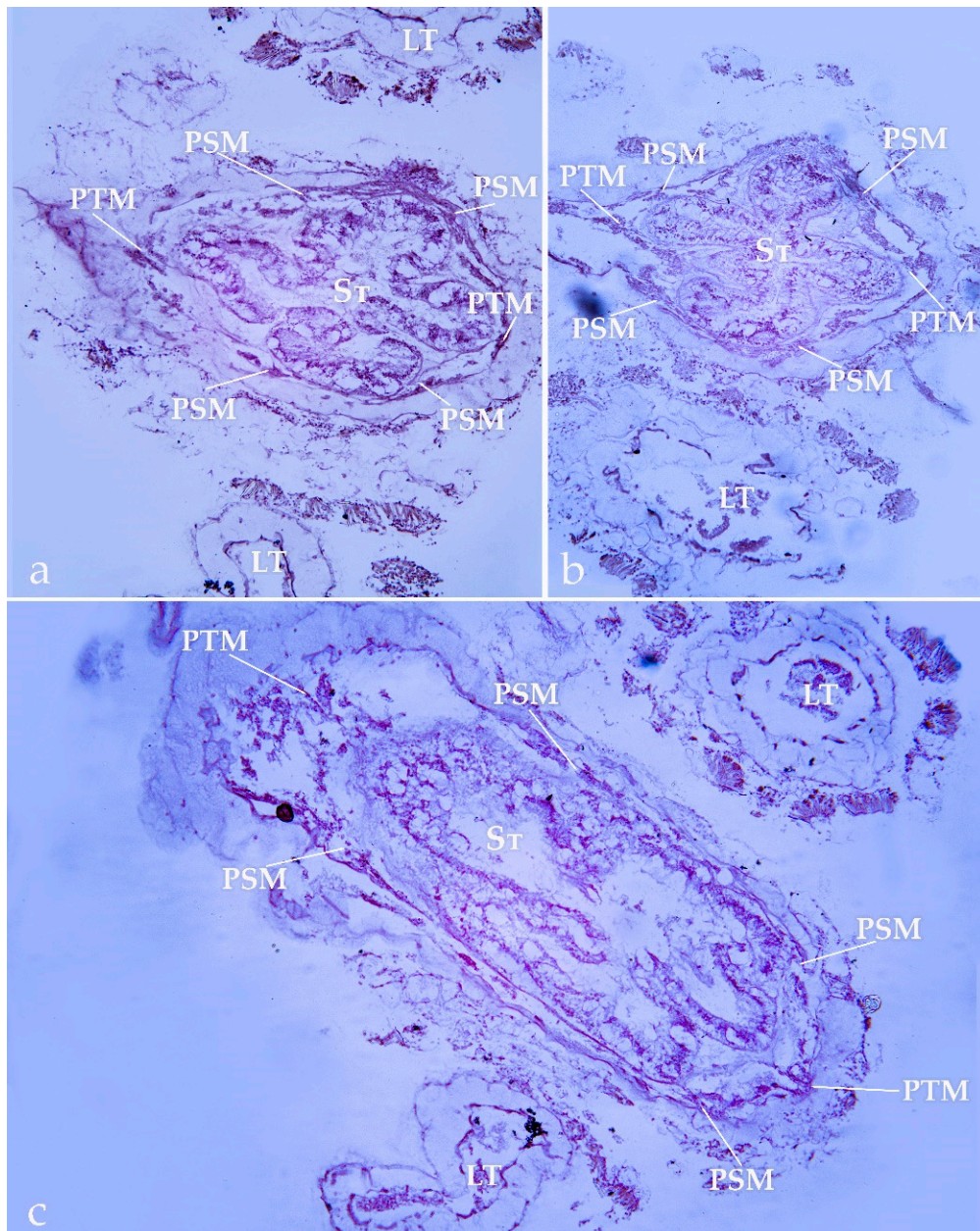

**Figure 5.** Histological section of polyps of *Abyssopathes lyra* (BPBM D2466). (**a**) cross-section of the oral cone near the base of the tentacles; (**b**) cross-section of the same polyp at the middle of the oral cone; (**c**) cross-section of another polyp at the middle of the oral cone. LT—lateral tentacle; St—stomodeum; PSM—primary sagittal mesentery; PTM—primary transversal mesentery. Scale 100 mkm.

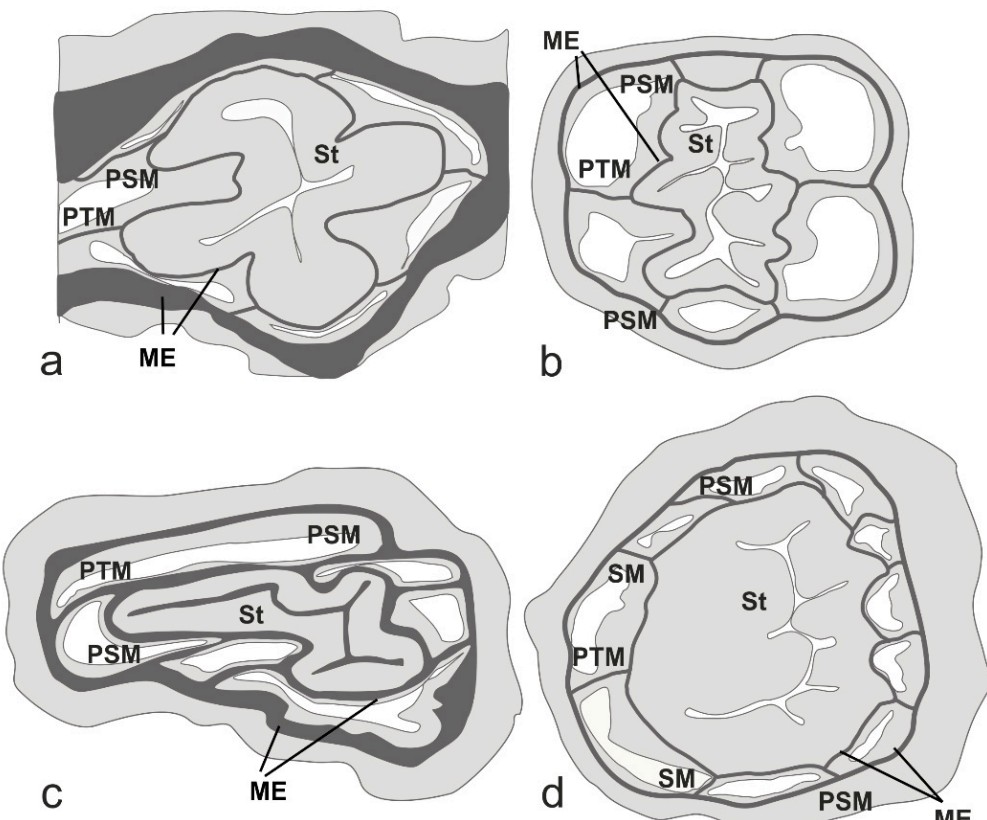

**Figure 6.** Schematic sections of the oral cone of polyps of Cladopathidae (**a–c**) and Schizopathidae (**d**). (**a**) *Abyssopathes lyra* (same section as shown in Figure 5a); (**b**) *Hexapathes heterosticha* Kinoshita, 1910 [30]; (**c**) *Cladopathes plumosa* [11]; (**d**) *Parantipathes* sp. (North Atlantic); St—stomodeum; PSM—primary sagittal mesentery; PTM—primary transversal mesentery; SM—secondary mesentery.

### 3.3. Molecular Analysis

A BLAST search in GenBank revealed that the *cox1* nucleotide sequence from *Abyssopathes lyra* (USNM 1453760) is a 99% match (Percent identity: 620/626) to the following taxa: *Heteropathes pacifica* (USNM 1234550, MBARI T886-A8), *Trissopathes pseudotristicha* Opresko, 2003 (USNM 1482134, ROV Jason II, Dive J2095-2-1-3, and USNM 98848, HAS-31), and *Chrysopathes formosa* Opresko, 2003 (USNM 1484087, ABL ID 41-99-6), all taxa in the family Cladopathidae.

Based on 626 bp of the mitochondrial *cox1* gene, genetic distances within the family Cladopathidae ranged from 0 to 1.95%. When comparing cladopathids to their nearest neighbor within the Schizopathidae (*Telopathes magna* MacIsaac and Best, 2013 [BAL103-1]), genetic distances ranged from 3.59 to 5.43%. However, the lower values were based on a comparison to *Trissopathes grasshoffi* Molodtsova, Altuna, and Hall-Spencer, 2019 (MT318862) and *Cladopathes* cf. *plumosa* (MT318861), which only had 167 bp and 258 bp, respectively. Not considering MT318862 and MT318861, the smallest genetic distance is 4.30%.

Results of the phylogenetic reconstruction for the families Cladopathidae and Schizopathidae are shown in Figure 7. *Abyssopathes lyra* groups in a clade containing other members of the family Cladopathidae. Within the Cladopathidae clade, which was recovered with 100% bootstrap support, *Heteropathes pacifica* (USNM 1234550), *Trissopathes pseudotristicha* (USNM 98848), and *Cladopathes* cf. *plumosa* (MT318861) are genetically identical. Grouping with this tri-generic complex is *Trissopathes* cf *tetracrada* Opresko, 2003 (MT318840), but it is genetically distinct with a K2P genetic distance of 0.16%. Three representative species of *Chrysopathes* (*C. formosa* [USNM 1015355, USNM 1484087 and NC_008411], *C. speciosa* [USNM 1484095] and *C.* cf. *micrantha* Opresko and Loiola, 2008 [MT318860]) are genetically distinct from one another and form a monophyletic clade. *Abyssopathes lyra* groups sister to the *Chrysopathes* clade,

although with little support (52). Based on genetic distance, *Trissopathes grasshoffi* (MT318862) appears to be closest to *Abyssopathes*; however, the *cox1* sequence for *T. grasshoffi* is only 167 bp in length, and thus this putative close relationship is likely an artifact of a lack of comparable sequence data.

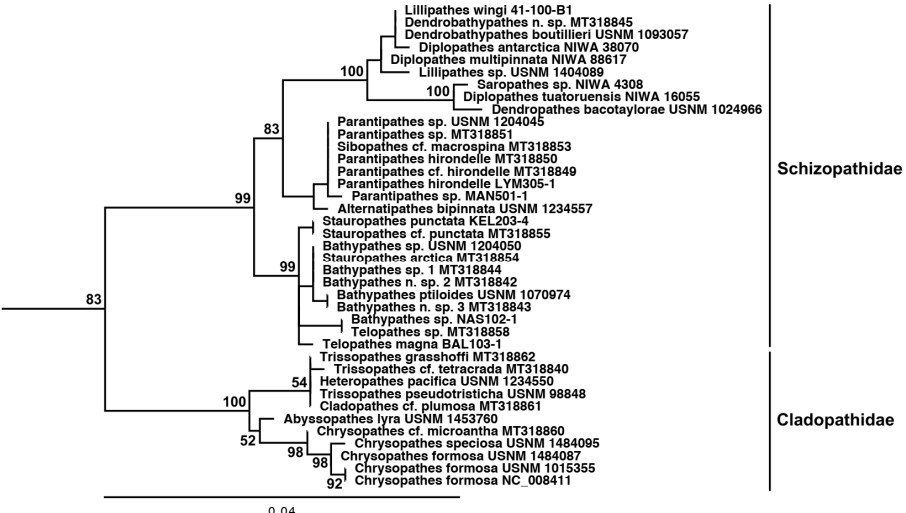

**Figure 7.** Partial results of a maximum likelihood-based phylogenetic reconstruction using mitochondrial *cox1* sequence data focusing specifically on the families Cladopathidae and Schizopathidae. The MAFFT LINS-i v7-based alignment consisted of 88 sequences and 626 sites. Akaike Information Criterion within jModelTest v2.1.1 selected the GTR+I+G model of sequence evolution (p-inv: 0.5690; gamma shape: 1.6810). IQ-Tree v2.1.2 utilized a BioNJ starting tree. Node support is based on 1000 ultrafast bootstrap replicates. The tree was rooted to sea anemones (*Metridium senile* (Linnaeus, 1761) and *Nematostella* sp.) and zoanthids (*Palythoa* sp. and *Savalia savaglia* (Bertoloni, 1819)). Following morphology-based taxonomic IDs are internal specimen codes followed by museum accession numbers (where applicable).

## 4. Discussion

Brook [11] had established the Schizopathinae for taxa whose polyps were said to be divided along the transverse axis into three sections, each bearing one pair of tentacles, which Brook regarded as "dimorphic zooids" (i.e., two gonozooids and one gastrozooid for each polyp). In *Schizopathes*, these divisions of a polyp were associated with external "peristomal involutions" (constrictions of the coenenchyme between the sagittal and each lateral pair of tentacles) and with internal "mesogloeal septa" hanging down from the interior surface of the coelenteron. All the genera placed in the subfamily Schizopathinae were considered by Brook to have "dimorphic" polyps with external "peristomal involutions" and internal "mesogloeal septa"; however, Brook [11] did not specifically describe these features for the polyps of *Bathypathes*, *Taxipathes*, or *Cladopathes*, and his illustrations suggest that, although the polyps of these genera were transversely elongated as in *Schizopathes*, they did not possess these features. Brook did note that the polyps of *Schizopathes* had ten mesenteries, six primary and four secondary mesenteries; however, the secondary mesenteries were found only in the upper part of the oral cone. Brook [11] implied that the same condition occurred in species of *Bathypathes*, although he did not specifically describe this condition in any species of *Bathypathes*, including *B. lyra*. Since then, ten distinctive mesenteries were reported in *Bathypathes arctica* (Lütken, 1871) [31], a species subsequently transferred to *Stauropathes* [12]. Later, Pasternak [32] reported six primary and four secondary mesenteries in *Bathypathes patula* from Kuril-Kamchatka Trench, but it is questionable if the material studied actually belonged to the genus *Bathypathes* [33]. On the other hand, only six primary mesenteries and no secondary mesenteries were found in the polyps of the type specimens of *Cladopathes plumosa* [11] (Figure 6c) and *Hexapathes*

*heterosticha* [30] (Figure 6b), and in addition, *C. plumosa* has a stomodeum similar in shape to that in *A. lyra*, a character not reported in any non-cladopathid genus. Therefore, the number of mesenteries and the shape of the stomodeum tie *Abyssopathes* more closely to the Cladopathidae than to the Schizopathidae.

Although the presence of only six mesenteries in the polyps has been used as a diagnostic feature of the family Cladopathidae, it has yet to be determined whether this feature is exclusive to the family. Despite the fact that only six mesenteries have been found in the polyps of the type material of both nominal species of *Sibopathes* [8,28], the position of this genus has not yet been confirmed with molecular data (see above). Specimens morphologically identified as *S. macrospina* have been shown to be genetically closely related to *Parantipathes* in the family Schizopathidae [6,7,9] (also see Figure 7), but these have not been evaluated histologically to determine the number of mesenteries in the polyps. If these specimens have only six mesenteries, as reported for the type species, it would indicate that this character state occurs in the Schizopathidae as well as the Cladopathidae. More histological studies of taxa of both the Schizopathidae and Cladopathidae are needed to determine if the character state of having only six mesenteries appeared multiple times and scattered between the two families.

The available genetic data clearly place *Abyssopathes lyra* in a clade containing only members of the family Cladopathidae. *Abyssopathes lyra* was found to group sister to the *Chrysopathes* clade, although with little support. Morphologically, the two genera are quite different. Species of *Chrysopathes* have primary pinnules arranged in six rows, whereas, in the other genera, including *Abyssopathes* (but excluding *Sibopathes* for reasons explained above), the primary pinnules occur in only three or four rows. As noted above, morphologically, *Abyssopathes* most closely resembles *Heteropathes* and *Hexapathes*. Additional genetic studies are needed to clarify the phylogenetic relationship of these taxa.

In conclusion, we consider the molecular and morphological evidence sufficient to transfer the genus *Abyssopathes* to the family Cladopathidae. After this taxonomic amendment, Cladopathidae becomes the second dominant family in the deep sea below 3500 m, with four species (29% of all known black corals) recorded from the abyssal and hadal zones.

**Supplementary Materials:** The following supporting information can be downloaded at: https://www.mdpi.com/article/10.3390/d15030436/s1, Table S1: Comparison of *Abyssopathes* Opresko, 2002 with other hitherto known genera of Cladopathidae and Schizopathidae.

**Author Contributions:** Conceptualization, T.N.M., D.M.O. and M.R.B.; methodology, D.M.O., M.R.B. and T.N.M.; morphological analysis D.M.O. and T.N.M., resources, D.M.O., T.N.M. and M.R.B.; molecular analysis M.R.B., M.O., Y.M.B., T.W.N. and R.F.R.; histological analysis T.N.M., U.V.S. and G.A.K.; writing—original draft preparation, D.M.O., T.N.M. and M.R.B.; writing—review and editing, D.M.O., T.N.M. and M.R.B.; supervision, M.R.B.; funding acquisition, D.M.O. and T.N.M. All authors have read and agreed to the published version of the manuscript.

**Funding:** Histological part of research was funded by RSF, grant number 22-24-00873 to TNM.

**Institutional Review Board Statement:** The study did not require ethical approval. Not applicable.

**Data Availability Statement:** *cox1* DNA sequence (GenBank Accession Number MT350286) obtained in the present study is submitted to the NCBI database and publicly available there (https://www.ncbi.nlm.nih.gov/nuccore/MT350286.1/ accessed on 30 December 2022).

**Acknowledgments:** The specimen on which this paper is based was collected by ROV Deep Discoverer during NOAA Cruise EX1703, "Discovering the Deep", to the Howland/Baker Unit of the Pacific Remote Islands Marine National Monument and the Phoenix Islands Protected Area on the R/V *Okeanos Explorer* under Permit Number PBRNo. 1/17 issued by the Republic of Kiribati. Thanks to the NOAA Office of Ocean Exploration and Research and the crew of the R/V *Okeanos Explorer*. Additional samples used in the molecular analysis were provided by R. Stone and J.F. Karinen of NOAA's Auk Bay Marine Laboratory, L. Londsten of the Monterey Bay Aquarium Research Institute, and S. France of the University of Louisiana, Lafayette. We would like to thank the Academic Editor and Reviewers for their constructive criticism and technical remarks that helped to improve this paper. Funding for DMO was provided, in part, by a grant from the U.S. Department of Justice to the

Smithsonian Institution. DMO and MRB are Research Associates at the Smithsonian Institution and gratefully acknowledge that affiliation.

**Conflicts of Interest:** The authors declare no conflict of interest. The funders had no role in the design of the study; in the collection, analyses, or interpretation of data; in the writing of the manuscript; or in the decision to publish the results.

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
