# Peer review of "One of the Deepest Genera of Antipatharia: Taxonomic Position Revealed and Revised"

_diversity, doi:10.3390/d15030436_

Round 1
Reviewer 1 Report
The authors have obtained a rarely collected species of an abyssal-dwelling black coral that has provided tissue for genetic and histological analysis. This has allowed reassessment of the systematic classification of the genus, which has further implications on the evolution of this group of cnidarians. While I believe the conclusions are sound, some clarifying points are needed throughout the manuscript; I have embedded comments in the PDF for authors to consider.

Author Response
We would like to thank the reviewer 1 for all the comments and remarks that allowed us to improve significantly our MS. Below we reply to particular questions raised by the reviewer 1
- . page 1 "Order Antipatharia is generally considered as primarily deep-sea with up to 75%" -"“deep sea” is only hyphenated when used as an adjective, so the hyphen is not appropriate here."
- We rephrased this line using more appropriate wording - Page 2 "which may be an adaptation to extremely deep env [5]" "Well, more accurately those authors put forth a conjecture (“Whether the loss of the COX1 intron is an adaptation to the deep-sea environment warrants further investigation.”), so this remains tenuous."
- We considered this comment and deleted mentioned text to avoid ambiguities. - page 2 "The genus Abyssopathes Opresko, 2002, comprises deep-sea black corals ..." - The title of the paper suggests Abyssopathes is the deepest dwelling black coral known, but there is no mention of that here. It would be an easy thing to say what depth the deepest specimens were collected from, and then put the colony sampled for this study into that context (perhaps it is the deepest one...?)
- We have re-phrased this text to make the wording more clear why we called the genus “the deepest” for the first time. However, we considered the remark of reviewer 1 and decided to modify the title using the wording “one of the deepest genera”. - page 4 - If this is to be the actual layout of the final paper, I would encourage trying to keep the figure legend on the same page as the figure, e.g. to make space move the text at the top of the page to the next page rather than shifting the figure legend.
- no, it was not intended layout. in our version Figures and corresponded Captions were at the same pages. We promise to check in the final version - page 4 - Spelling error: Schizopathidae
fixed - page 4 - Table 1
Shouldn’t a “maximum number” be one value? Several of these show a range. Why is the maximum not being shown? Should this parameter be named something different?
we considered this comment and changed “maximum number” to “typical number” - Page 7 - Same comment as for Fig 3 legend
- no, it was not intended layout. in our version Figures and corresponded Captions were at the same pages. We promise to check in the final version - Page 7 - "are relatively short in proportion to the transverse polyp diameter
This is presented as quite anecdotal and based on only a couple of photographs from each taxon. Were any measurements taken of the tentacles vs polyp transverse diameter? Were the observations from only a single branch or colony, etc? Have any other authors explored this quantitatively?
Unfortunately the only answer to all questions set by reviewer 1 above is “no”. There were no any estimation done. All our observations were done at in situ photographs (not all done in good projection) – as were mentioned in the text. We decided to rephrase this part of the MS to make all our observations and statements clearer. - Figure 5
This is a critical figure relative to the argument being made in this paper since it purportedly shows the characteristic 6 mesenteries (I confess I can’t seem them, but I have little expertise with such image data) and so the figure should be reproduced larger on the page or oriented sideways on a full page.
I consulted with a colleague who is an expert in histology and he remarked that these were of relatively low quality for cnidarians, likely a factor out of the authors control and based on the initial preservation of the sample (in ethanol directly, I presume, and not in another fixative such as formalin or glutaraldehyde - but it would be useful to provide those details in the Methods so readers understand the limitations). I think it would be helpful if the images were accompanied by a line drawing sketch that showed the key characteristics.
- We took very seriously this comment by the reviewer 1. (1) We re-formatted Figure 5 to provide better view of histological sections and (2) added new Figure 6 with schematic outlines of histological sections of several species of Schizopathidae and Cladopathidae (two from literature and two based on original data). We have to note here that all data on fixative used was already provided in Material and Methods and Results (see e.g. page 7), but we modified this part of the MS to make this information (and limitations) more visible for the reader. - Page 8 “with little support (48)“ i) if 48 is meant to be a support value, why does Fig 6 show 52 at this node?ii) I would argue such a support value means NO support, e.g. it suggests more than 50% of the time this relationship is not recovered
- The bootstrap value of 48% was based on an older version of the phylogenetic tree. We failed to update the number. Apologies. Thank you for pointing this out to us. After building the initial tree, we decided to include cox1 sequences from Barrett et al (2020) as well as the new Diplopathes cox1 sequences from Opresko et al (2022). After including the additional taxa, the bootstrap value increased from 48 to 52. Per Hillis & Bull (1993), clades are considered supported if they have bootstrap support estimates equal to or greater than 70%. We fully acknowledge that this node (52%) has "little support." - Page 9 “subsequently"
done - Page 9 “Despite that the presence…”
done - Page 9 “confirmed”
done - Page 9 I think this requires further explanation. There has been no discussion of the position of Sibopathes in the phylogeny prior to this comment (in the introduction reference was made to previous studies that suggest the two species of Sibopathes may belong to different genera, although in all cases no genetic data has yet been obtained from the type species, so this is actually unknown). It is misleading to say that the position is not confirmed. The tree in Fig 6 shows the available Sibopathes sequence to be solidly among Parantipathes [in fact, possibly identical for this cox1 region], as does the more extensive dataset in Barrett et al. What is at question is whether that particular specimen has only 6 mesenteries (apparently only the specimens from the Gulf of Mexico have been examined histologically, and they had 6 mesenteries, so the assumption has carried that that applies to all the cf macrospina) and how it relates to the type species.
The data already show this to be the case. cf macrospina is well-supported in a different clade. So if you believe the data that that species has only primary mesenteries, then you know this character is not restricted to a single clade. I think you are confusing the question of the monophyly of Sibopathes with the distribution of this character state
- As it was mentioned in the introduction, there is no DNA data for the type of Sibopathes gephura, the type species of the genus. Also, no DNA data was obtained for the type material of S. macrospina. The genetic affinity to Parantipathes was reported for specimens identified as S.macrospina basing on gross morphology only. We did our best to rephrase this part of the MS to make all this information more clear.
- Data Availability Statement:
In virtually all cases that I am familiar with, a journal would require that new sequence data be accompanied by the GenBank accession numbers
- The GenBank accession number was indicated in the MS. However we admit that we did it rather briefly in the text of the Material and Methods section. We would like to thank reviewer 1 for pointing it out. We did our best to make this statement more clear in Data Availability Statement
Reviewer 2 Report
Order Antipatharia is generally considered as primarily deep-sea with up to 75% of all hitherto known antipatharian species found below 50 m [1,2]
Such statement is constantly used although is based on those two ‘old’ references. Even when considering another more recent review (Bo et al. 2019 - Antipatharians of the Mesophotic Zone: Four Case Studies) - the current diversity and abundance of antipatharians known is bias by the higher number of studies of this taxa derived from deep-sea surveys. Little consideration has been given to the diversity and abundance of antipatharians above 50 m depth.
It should be noted that Abyssopathes anomala is uniquely different from the other two species of Abyssopathes in that it lacks anterior primary pinnules, a condition which may be secondarily derived.
What do you mean by secondarily derived?
Cross section of the upper part of the stomatodeum have very characteristic Maltese cross or four-leaf outline (Figure 5 b and c) never reported in other antipatharians. The cross becomes more elongated in transverse direction near the bases of the tentacles (Figure 5a), but retains a distinct four-leaf shape.
It would be very useful to include an image of another species/specimen to be able to show the difference. Additionally, it might be adequate to acknowledge that you are only examining one individual and that such characteristics might or might not be the norm of the species.
This 626 bp fragment of cox1 was added to the igrC multiple sequence alignment from [4]. Additionally, schizopathid and cladopathid mitochondrial cox1 sequences were extracted from the complete mitogenome-based phylogeny presented in [4]
What is the reason only one gene (cox1) was used in this study? At least one nuclear gene would be good (ITS1 used in various other antipatharian studies), which is available from Ref [4] for some the species used for comparison in this study.
Despite the presence of only six mesenteries was conformed for the both species of Sibopathes [6,25], position of this genus was not confirmed by molecular data [4,5,7] (see above). More histological studies of hitherto known members of Schizopathidae and Cladopathidae is needed to reveal, if the condition with only primary mesenteries appeared several times and scattered between two families.
Needs ‘editorial’ revision.
Therefore, further study may support the conclusion that the number of rows of pinnules is a more appropriate basis for subdividing the family into subfamilies.
Do you mean this study supports that conclusion? Also, it is a more appropriate factor (basis) compared to?
A general editorial revision needed. A glossary is needed.
Author Response
We would like to thank the reviewer 2 for the criticism, with comments and remarks provided we did our best to correct the MS.
- Order Antipatharia is generally considered as primarily deep-sea with up to 75% of all hitherto known antipatharian species found below 50 m [1,2]
Such statement is constantly used although is based on those two ‘old’ references. Even when considering another more recent review (Bo et al. 2019 - Antipatharians of the Mesophotic Zone: Four Case Studies) - the current diversity and abundance of antipatharians known is bias by the higher number of studies of this taxa derived from deep-sea surveys. Little consideration has been given to the diversity and abundance of antipatharians above 50 m depth.
- We agree well, that this approximation is a bit outdated. So we have rephrased this part of Introduction to encompass Bo et all., 2019 approximations of black coral diversity at mesophotic depths. But to be honest in our MS we aim to the extreme deep sea and do not discuss much shallow-water fauna. - It should be noted that Abyssopathes anomala is uniquely different from the other two species of Abyssopathes in that it lacks anterior primary pinnules, a condition which may be secondarily derived.
What do you mean by secondarily derived?
- We have rephrased this part of the Introduction as following: “The presence of anterior pinnules is a characteristic of all other taxa within the family Cladopathidae, and its absence in A. anomala suggests that it is an autapomorphic character state.” - Cross section of the upper part of the stomatodeum have very characteristic Maltese cross or four-leaf outline (Figure 5 b and c) never reported in other antipatharians. The cross becomes more elongated in transverse direction near the bases of the tentacles (Figure 5a), but retains a distinct four-leaf shape.
It would be very useful to include an image of another species/specimen to be able to show the difference. Additionally, it might be adequate to acknowledge that you are only examining one individual and that such characteristics might or might not be the norm of the species.
We have added sketch outline of several other species (Figure 6), that include cladopathids and one schizopathid. As for acknowledgement of one individual study, we are just reporting what we have seen at histological sections. On the other hand (1) we did section of at least two polyps, both presented at Figure 5, (2) TM and DO have seen a similar shape in another specimen of Abyssopathes lyra collected in CCZ (Molodtsova, Opresko, 2017), so we prefer not to put this statement. - This 626 bp fragment of cox1 was added to the igrC multiple sequence alignment from [4]. Additionally, schizopathid and cladopathid mitochondrial cox1 sequences were extracted from the complete mitogenome-based phylogeny presented in [4]
What is the reason only one gene (cox1) was used in this study? At least one nuclear gene would be good (ITS1 used in various other antipatharian studies), which is available from Ref [4] for some the species used for comparison in this study.
We agree that more data is always better, especially if those data come from an additional source; i.e., the nuclear genome. A subsample of the specimen under consideration is housed at the Smithsonian Institution's National Museum of Natural History in Washington DC, which is where the molecular analysis took place. More specifically, the DNA was extracted, amplified and sequenced at the Laboratories of Analytical Biology (LAB). The LAB is the biotechnology core of the NMNH molecular research programs. LAB is affiliated with the Consortium for the Barcode of Life as well as the Barcode of Life Database. The LAB primarily sequences the CO1 Barcode of Life, which is the gene that was analyzed herein. Coauthor M. Brugler was previously employed at the AMNH in NYC where he had an in-house ABI-3730xL Sanger Sequencer and thus could supplement the traditional genes (cox1) with other barcodes, both from the mitochondrial and nuclear genome. However, M. Brugler transitioned to a teaching institution in 2020 and no longer has funding for, or access to, Sanger sequencing. Thus, this particular team of coauthors had to rely upon the LAB and their standard barcode. Co-author M. Brugler cloned ITS1 and ITS2 in black corals during his PhD and found multiple copies within a single individual. Thus, it is the opinion of co-author M. Brugler that ITS is not a suitable molecular marker unless the PCR products are cloned or if next-gen amplicon sequencing is conducted. While Phase Determination is often recommended (see Flot, J.F., Tillier, A., Samadi, S. and Tillier, S., 2006. Phase determination from direct sequencing of length‐variable DNA regions. Molecular Ecology Notes, 6(3), pp.627-630.), black corals appear to have more than two copies of ITS per individual and thus this method is not suitable. Rest assured that SI NMNH postdoc Dr. Jeremy Horowitz is currently sequencing UCEs and exonic loci for all species of black coral within the SI NMNH collection. - Despite the presence of only six mesenteries was conformed for the both species of Sibopathes [6,25], position of this genus was not confirmed by molecular data [4,5,7] (see above). More histological studies of hitherto known members of Schizopathidae and Cladopathidae is needed to reveal, if the condition with only primary mesenteries appeared several times and scattered between two families.
Needs ‘editorial’ revision.
We have revised this part of the MS in response to the comments of reviewers 1 and 2 to make all of our statements clearer and more comprehensible to the reader. - Therefore, further study may support the conclusion that the number of rows of pinnules is a more appropriate basis for subdividing the family into subfamilies.
Do you mean this study supports that conclusion? Also, it is a more appropriate factor (basis) compared to?
This part was retained since the first version of the MS that did not include histology section. Thank you for catching it up. We carefully revised and reworded Discussion section to avoid any ambiguities. - A general editorial revision needed.
We did our best to review the MS according to all the comments and recommendations from reviewers 1 and 2 and doublechecked for any inconsistency or bad wording.
8. A glossary is needed.
We believe that all of the terminology used in the MS is commonly used for the group; adding a glossary would significantly and unnecessarily increase the volume of MS. Instead, we would like to draw readers' attention to two relatively recent papers that already have extensive terminology sections: Brugler et al., 2013 and Opresko et al., 2014.